# Action Mapping for Reinforcement Learning in Continuous Environments with Constraints

## Abstract

Deep reinforcement learning (DRL) has had success across various domains, but applying it to environments with constraints remains challenging due to poor sample efficiency and slow convergence. Recent literature explored incorporating model knowledge to mitigate these problems, particularly through the use of models that assess the feasibility of proposed actions. However, integrating feasibility models efficiently into DRL pipelines in environments with continuous action spaces is non-trivial. We propose a novel DRL training strategy utilizing *action mapping* that leverages feasibility models to streamline the learning process. By decoupling the learning of feasible actions from policy optimization, action mapping allows DRL agents to focus on selecting the optimal action from a reduced feasible action set. We demonstrate through experiments that action mapping significantly improves training performance in constrained environments with continuous action spaces, especially with imperfect feasibility models.

## 1 Introduction

Deep Reinforcement learning (DRL) has emerged as a powerful tool across numerous application domains, ranging from robotics (Funk et al., 2022) and autonomous system (Trumpp et al., 2024) to game-playing (Vinyals et al., 2019) and other decision-making tasks Bayerlein et al. (2021). The ability of DRL to learn complex behaviors through trial and error makes it highly promising for solving challenging problems. Despite the potential of DRL, its application is limited by poor sample efficiency and challenges in handling constraints that frequently arise in real-world tasks. Such constraints, such as safety requirements or physical limits, complicate the exploration and learning processes (Achiam et al., 2017).

In many constrained environments, it is essential to prevent the agent from selecting actions that could violate certain constraints. Therefore, it has been proposed to utilize feasibility models that assess the feasibility of proposed actions at a given state. Their application is straightforward in discrete action spaces, where infeasible actions can simply be masked out, preventing the agent from choosing them (Huang & Ontañón, 2020). Action masking has been successfully applied in various discrete action space problems, such as vehicle routing (Nazari et al., 2018), autonomous driving scenarios (Krasowski et al., 2020), and task scheduling (Sun et al., 2024). However, integrating feasibility models becomes significantly more challenging in continuous action spaces, as actions cannot be directly masked out.

Several methods have been proposed for integrating feasibility models in continuous action spaces. *Action replacement* substitutes infeasible actions with predefined feasible ones (Srinivasan et al., 2020), while *action resampling* rejects invalid actions and samples new ones until a feasible action is found (Bharadhwaj et al., 2020). Another approach is *action projection*, which projects the agent's chosen action onto the nearest feasible one (Cheng et al., 2019). While these methods offer solutions, they often introduce inefficiencies or increase the computational cost of learning and decision-making, particularly in complex environments.

Inspired by the simplicity of action masking in discrete action spaces, Theile et al. (2024) proposed *action mapping*, a framework to address inefficiencies when incorporating feasibility models in continuous action spaces. In their framework, a feasibility policy is pretrained to generate all feasible actions by leveraging the feasibility model. This feasibility policy creates a state-dependent representation of feasible actions, enabling an objective policy to focus solely on optimizing the task's

objective. While this concept shows promise, Theile et al. (2024) only focussed on the feasibility policy, omitting the objective policy and thus leaving its practical benefits in DRL unexplored. In this paper, we refine their feasibility policy training and formulate the training procedure for the objective policy. We further demonstrate its implementation in safe RL, achieving significant improvements in both sample efficiency and constraint satisfaction compared to other safe RL methods.

Importantly, our focus is not on ensuring guaranteed constraint satisfaction. Instead, the primary objective is to leverage prior knowledge encapsulated in (potentially imperfect) feasibility models as an inductive bias to accelerate and improve the performance of DRL training. To that end, we test our approach in two constrained environments: (i) A robotic arm end-effector pose positioning task with obstacles using a perfect feasibility model, and (ii) a path planning environment with constant velocity and non-holonomic constraints, for which an approximate feasibility model is used. The experiments show that our action mapping approach outperforms Lagrangian approaches (Ha et al., 2020; Ray et al., 2019) and an action projection approach, especially in scenarios with an approximate feasibility model.

Our work bridges the gap between the conceptual framework of action mapping and its practical implementation in constrained environments, leading to the following contributions:

- Development and implementation of the action mapping framework for DRL to efficiently incorporate feasibility models during training.
- Demonstration of action mapping's (AM) effectiveness in constrained environments using perfect and approximate feasibility models with AM-PPO and AM-SAC implementations.
- Empirical comparison with Lagrangian methods and action replacement, resampling, and projection, highlighting superior performance, especially with approximate models.
- Showcasing action mapping's ability to express multi-modal action distributions, enhancing exploration and learning performance.

## 2 PRELIMINARIES

### 2.1 STATE-WISE CONSTRAINED MARKOV DECISION PROCESS

A state-wise constrained Markov Decision Process (SCMDP) (Zhao et al., 2023) can be defined through the tuple $(\mathcal{S}, \mathcal{A}, \mathrm{R}, \{\mathrm{C}_i\}_{\forall i}, \mathrm{P}, \gamma, \mathcal{S}_0, \mu)$, in which $\mathcal{S}$ and $\mathcal{A}$ are the state and action space. The reward function $\mathrm{R} : \mathcal{S} \times \mathcal{A} \to \mathbb{R}$ defines the immediate reward received for performing a specific action in a given state. A transition function $\mathrm{P} : \mathcal{S} \times \mathcal{A} \to \mathcal{P}(\mathcal{S})$ describes the stochastic evolution of the system, with $\mathcal{P}(\mathcal{S})$ defining a probability distribution over the state space. The discount factor $\gamma \in [0, 1]$ weighs the importance of immediate and future rewards. Additionally, a set of initial states $\mathcal{S}_0$ and an initial state distribution $\mu = \mathcal{P}(\mathcal{S}_0)$ are provided. When following a stochastic policy $\pi : \mathcal{S} \to \mathcal{P}(\mathcal{A})$, the expected discounted cumulative reward is defined as

$$\mathrm{J}(\pi) = \mathbb{E}\left[\sum_{t=0}^{\infty} \gamma^t \mathrm{R}(s_t, a_t) \mid s_0 \sim \mu, a_t \sim \pi(\cdot|s_t), s_{t+1} \sim \mathrm{P}(\cdot|s_t, a_t)\right]. \tag{1}$$

In contrast to a regular MDP (Sutton, 2018), in an SCMDP, a set of cost functions $\{\mathrm{C}_i\}_{\forall i}$ is defined, in which $\mathrm{C}_i : \mathcal{S} \times \mathcal{A} \times \mathcal{S} \to \mathbb{R}$, such that every transition is associated with a cost value. In a CMDP (Altman, 2021), the expected discounted cumulative cost for each cost function $\mathrm{C}_i$ needs to be bounded by a $w_i \in \mathbb{R}$. In an SCMDP, the cost functions are required to be bounded for *each* transition individually, which is a stricter constraint. With all possible trajectories $\tau^\pi(s)$ when following $\pi$ starting from a state $s$, the SCMDP optimization problem is formulated as

$$\pi^* = \arg\max_{\pi \in \Pi} \mathrm{J}(\pi)$$
$$\text{s.t. } \mathrm{C}_i(s_t, a_t, s_{t+1}) \le w_i, \quad \forall i, \forall (s_t, a_t, s_{t+1}) \sim \tau^\pi(s_0), \forall s_0 \in \mathcal{S}_0. \tag{2}$$

The optimization requires that each cost function $\mathrm{C}_i$ is bounded by $w_i$ for each transition $(s_t, a_t, s_{t+1}) \sim \tau^\pi(s_0)$ along all possible trajectories of $\pi$ starting from all possible initial states in $\mathcal{S}_0$.

## 2.2 DEEP REINFORCEMENT LEARNING

In reinforcement learning, the goal is to learn a policy $\pi(a|s)$ that maximizes the expected cumulative reward as in equation 1 (Sutton, 2018). In DRL, policies are parameterized by deep neural networks, enabling agents to handle high-dimensional state and action spaces. For environments with continuous action spaces, two widely-used DRL algorithms are the off-policy Soft Actor-Critic (SAC) (Haarnoja et al., 2018) and on-policy Proximal Policy Optimization (PPO) (Schulman et al., 2017) algorithms.

**Soft Actor-Critic (SAC).** SAC is an off-policy algorithm that maximizes a reward signal while encouraging exploration through an entropy regularization term. The policy $\pi$ in SAC aims to maximize both the expected cumulative reward and the policy entropy, encouraging stochasticity in action selection. The SAC objective is defined as

$$J(\pi) = \mathbb{E}_{(s_t, a_t) \sim \pi} \left[ Q(s_t, a_t) + \alpha \mathcal{H}(\pi(\cdot|s_t)) \right], \tag{3}$$

where $Q(s_t, a_t)$ represents the Q-function, which estimates the expected return when taking action $a_t$ in state $s_t$, and $\mathcal{H}(\pi(\cdot|s_t))$ is the entropy of the policy. The hyperparameter $\alpha$ controls the trade-off between return maximization and exploration.

**Proximal Policy Optimization (PPO).** PPO is an on-policy algorithm that improves training stability by limiting the magnitude of policy updates to ensure smoother learning. The PPO objective is defined using a clipped surrogate loss to avoid large deviations from the current policy:

$$J(\pi) = \mathbb{E}_t \left[ \min \left( r_t(\pi) \hat{A}_t, \text{clip}(r_t(\pi), 1 - \epsilon, 1 + \epsilon) \hat{A}_t \right) \right], \tag{4}$$

where $r_t(\pi)$ is the probability ratio between the new and old policies, and $\hat{A}_t \approx Q(s_t, a_t) - V^\pi(s_t)$ is the approximate advantage function. In the advantage, the state value $V^\pi$ is the expected value if following the current policy. The advantage is commonly estimated using the generalized advantage estimate (GAE) from Schulman et al. (2015).

## 2.3 FEASIBILITY MODELS

Given the individual cost functions for a transition, a joint cost function can be defined as

$$C(s_t, a_t, s_{t+1}) = \sum_{\forall i} \max \left\{ 0; \ C_i(s_t, a_t, s_{t+1}) - w_i \right\}, \tag{5}$$

which is 0 if no cost function exceeds its bound and otherwise the sum of the violations. With the joint cost function, a policy-dependent trajectory cost can be defined as

$$C^\tau(s; \pi) = \max_{(s_t, a_t, s_{t+1}) \sim \tau^\pi(s)} C(s_t, a_t, s_{t+1}) \tag{6}$$

that expresses the highest joint cost of any transition $(s_t, a_t, s_{t+1})$ along all possible trajectories, starting from some $s$ and following $\pi$. It is 0 if no cost function exceeds its bound.

A feasibility model $G : \mathcal{S} \times \mathcal{A} \to \mathbb{R}$ defines the cost violation of the transition induced by applying action $a_t$ at state $s_t$, plus the cost violation of the most feasible policy from the next state. Formally, we define it as

$$G(s_t, a_t) = \max_{s_{t+1} \sim P(\cdot|s_t, a_t)} \left[ C(s_t, a_t, s_{t+1}) + \min_{\pi \in \Pi} C^\tau(s_{t+1}; \pi) \right], \tag{7}$$

with the maximization over all possible next states. A Boolean version of the feasibility model $g : \mathcal{S} \times \mathcal{A} \to \mathbb{B}$ can be defined as

$$g(s_t, a_t) = (G(s_t, a_t) == 0), \tag{8}$$

where "==" denotes Boolean equality. It indicates whether all possible transitions induced by $a_t$ at $s_t$ are feasible and whether a policy exists such that a cost violation can be avoided from any possible $s_{t+1}$.

## 3 RELATED RESEARCH

The majority of the safe RL literature focuses on discrete actions through action masking (Huang & Ontañón, 2020), often through shielding (Alshiekh et al., 2018). Action projection is usually proposed to incorporate feasibility models for continuous action spaces. Donti et al. (2021) propose DC3 to perform action projection through gradient descent on a feasibility model, while Cheng et al. (2019) use control barrier functions. When the feasible action space is known and can be described as a convex polytope, a limiting assumption, Stolz et al. (2024) introduce a similar concept to action mapping that also improves performance.

Learning-based approaches incorporate constraints directly into the RL process, often using techniques like Lagrangian optimization or dual frameworks. CMDPs (Altman, 2021) introduce cumulative cost constraints, while Constrained Policy Optimization (CPO) Achiam et al. (2017) extends trust-region policy optimization by ensuring monotonic improvement under safety constraints. Bharadhwaj et al. (2020) propose conservative safety critics that reject unsafe actions during exploration and resamples from the actor, while Srinivasan et al. (2020) replaces the unsafe action with a null action. Penalized PPO (P30) (Zhang et al., 2022) further incorporates constraint penalties, and Lagrangian PPO (Ray et al., 2019) and feasible actor-critic (Ma et al., 2021) use Lagrangian multipliers to balance reward maximization and constraint satisfaction.

Combining model-based and learning approaches, learned feasibility models are often used to perform action projection. Dalal et al. (2018) learn a linear approximation of the cost function and perform action projection using the linear model. Chow et al. (2019) learn a Lyapunov function and perform action or parameter projection. Zhang et al. (2023) use DC3 with a learned safety critic, using it for iterative gradient descent-based action projection. Further approaches to safe RL can be found in surveys by Gu et al. (2022b) and Zhao et al. (2023).

While recent works have primarily focused on action projection methods when incorporating feasibility models, we propose a novel method that describes a mapping instead of a projection, which we show can improve learning performance.

Besides the safe RL perspective, action mapping is also related to the research in action representation learning. In action representation, a continuous latent action representation of large (discrete) action spaces is learned (Chandak et al., 2019). It has been extended to learn representation of sequences of actions (Whitney et al., 2020), mixed discrete and continuous actions (Li et al., 2022), or specifically for offline RL (Gu et al., 2022a). Action mapping can be thought of as finding an action representation for the state-dependent set of feasible actions.

## 4 ACTION MAPPING METHODOLOGY

If a feasibility model g from equation 8 can be derived for an environment, the question is how to use it efficiently in RL. The intuition of *action mapping* is to first learn *all* feasible actions through interactions with g and subsequently train a policy through interactions with the environment to choose the best action among the feasible ones. By allowing the objective policy to choose only among feasible actions, the SCMDP is effectively transformed into an unconstrained MDP, as illustrated in Fig. 1. Since unconstrained MDPs are generally easier to solve than SCMDPs–primarily due to reduced exploration complexity–action mapping can drastically improve training performance. In the following, the time index is dropped from $s_t$ and $a_t$ for improved readability.

Formally, given g, the state-dependent set of feasible actions $\mathcal{A}_s^+ \subseteq \mathcal{A}$ contains all actions for which $g(s, a) = 1$. To learn all feasible actions, a feasibility policy is defined as

$$\pi_f : \mathcal{S} \times \mathcal{Z} \to \mathcal{A}_s^+, \tag{9}$$

which is a generator that generates feasible actions for a given state. The latent space $\mathcal{Z}$ allows the policy $\pi_f$ to generate multiple feasible actions for the same state. It has the same cardinality as the action space $\mathcal{A}$. A perfect feasibility policy is a state-dependent surjective map from $\mathcal{Z}$ to $\mathcal{A}_s^+$, as it is able to generate all feasible actions without generating infeasible ones. With a perfect feasibility policy, the latent space $\mathcal{Z}$ is an action representation of the set of feasible actions.

Given a feasibility policy $\pi_f$, an objective policy

$$\pi_o : \mathcal{S} \to \mathcal{P}(\mathcal{Z}) \tag{10}$$

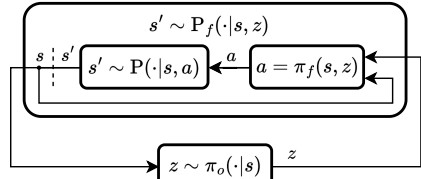

Figure 1: Interaction architecture of action mapping, showing how a perfect feasibility policy $\pi_f$ can transform the SCMDP indicated by P into an unconstrained MDP with transition function $P_f$ and action space $\mathcal{Z}$.

can be trained to find the state-dependent optimal latent value distribution. Through the map $\pi_f$, the overall policy

$$\pi = \pi_f \circ \pi_o : \mathcal{S} \to \mathcal{P}(\mathcal{A}_s^+) \tag{11}$$

thus learns the optimal distribution over the feasible actions.

Fig. 1 shows the action mapping architecture. It illustrates that a perfect feasibility policy $\pi_f$ transforms the state-wise constrained environment with transition function P into an unconstrained environment with transition function $P_f$ with action space $\mathcal{Z}$. The following describes how to train the feasibility and objective policies.

### 4.1 FEASIBILITY POLICY

To train the feasibility policy, we use the approach from Theile et al. (2024). The parameterized feasibility policy $\pi_f^\theta$ with parameters $\theta$ aims to be a state-dependent surjective map from the latent space $\mathcal{Z}$ to the set of feasible actions $\mathcal{A}_s^+$, to allow the objective policy to choose among all feasible actions. When sampling $z \sim \mathcal{U}(\mathcal{Z})$, i.e., uniformly from the latent space, $\pi_f^\theta$ becomes a generator with a conditional probability density function (pdf) $q^\theta(a|s)$. Since $\pi_f^\theta$ is task-independent, this generator should generate all feasible actions equally likely without any bias toward any specific feasible action. Therefore, the target of $q^\theta(a|s)$ is a uniform distribution in the feasible action space, i.e., $\mathcal{U}(\mathcal{A}_s^+)$.

This uniform target distribution is given through the feasibility model g as

$$p(a|s) = \frac{g(s,a)}{Z(s)}, \quad \text{with} \quad Z(s) = \int_{\mathcal{A}} g(s, a') da' \tag{12}$$

where $Z(s)$ is a partition function, effectively indicating the *volume* of feasible actions given a state. The objective of the feasibility policy is then $\min_\theta D(p(\cdot|s) || q^\theta(\cdot|s))$, with a divergence measure D.

Since $q^\theta$ and $p$ are not available in closed form, both need to be approximated. The distribution of the policy can be approximated through a kernel density estimate (KDE) based on $N$ samples from $q^\theta(\cdot|s)$ as

$$\hat{q}_\sigma^\theta(a|s) = \frac{1}{N} \sum_{a_i \sim q^\theta(\cdot|s)} k_\sigma(a - a_i), \tag{13}$$

with a kernel $k$ with bandwidth $\sigma$. To explore the feasibility of actions outside the support of $q^\theta$, Gaussian noise is added to the sampled actions as

$$a_i^* = a_i + \epsilon_i, \quad \epsilon_i \sim \mathcal{N}(0, \sigma'), \tag{14}$$

which is equivalent to sampling from the KDE with bandwidth $\sigma'$ as $a_i^* \sim \hat{q}_{\sigma'}^\theta(\cdot|s)$. Theile et al. (2024) propose to sample multiple actions per support point of the KDE, which our experiments showed to be unnecessary. The distribution $\hat{q}_{\sigma'}^\theta$ with $\sigma' \geq \sigma$ is a proposal distribution used for the divergence estimate. Using these samples, the target distribution can be estimated as

$$\hat{p}(a|s) = \frac{g(s,a)}{\hat{Z}(s)}, \quad \text{with} \quad \hat{Z}(s) = \frac{1}{N} \sum_{a_i^* \sim \hat{q}_{\sigma'}^\theta(\cdot|s)} \frac{g(s, a_i^*)}{\hat{q}_{\sigma'}^\theta(a_i^*|s)}, \tag{15}$$

where the partition function is approximated using Monte-Carlo importance sampling. Using the approximation $\hat{q}^{\theta}_{\sigma}$, the samples $a^*_i$ from the proposal distribution $\hat{q}^{\theta}_{\sigma'}$, and the approximation of the target distribution $\hat{p}$, the gradient of the Jensen-Shannon divergence can be approximated as

$$\frac{\partial}{\partial \theta} D_{JS}(p \,||\, q^{\theta}) \approx \frac{1}{2N} \sum_{a^*_i \sim \hat{q}^{\theta}_{\sigma'}} \frac{\hat{q}^{\theta}_{\sigma}(a^*_i)}{\hat{q}^{\theta}_{\sigma'}(a^*_i)} \log\left(\frac{2\hat{q}^{\theta}_{\sigma}(a^*_i)}{\hat{p}(a^*_i) + \hat{q}^{\theta}_{\sigma}(a^*_i)}\right) \frac{\partial}{\partial \theta} \log \hat{q}^{\theta}_{\sigma}(a^*_i), \qquad (16)$$

dropping the dependency on $s$ for readability. Theile et al. (2024) showed the Jensen-Shannon divergence yields the best compromise between reaching all actions, even in disconnected sets of feasible actions, and minimizing the probability of generating infeasible actions. An algorithm and implementation details in the context of DRL are provided in Appendix A.

## 4.2 OBJECTIVE POLICY

Algorithm 1 shows our proposed training procedure for AM-SAC based on Soft Actor-Critc (SAC) and AM-PPO using Proximal Policy Optimization (PPO). For both algorithms, the objective policy $\pi^{\phi}_o$ parameterizes a Gaussian from which a value $x$ is sampled (lines 6-7). Since the Gaussian is not bounded, we squash it using a $\tanh$ yielding the latent value $z$ (line 8). The latent $z$ is then mapped to an action $a$ using $\pi^{\phi}_f$ (line 9). If using AM-PPO, the state-value estimate $\hat{V}_s$ and the log-likelihood of the latent are gathered (lines 10-11). For the log-likelihood, a term is added to compensate for the squashing effect. Using the action $a$, the environment steps to the next state $s'$, yielding reward $r$ and a termination flag $d$ (line 12), and the collected experience tuples are stored in the buffer $\mathcal{D}$ (line 13). The experience tuples do not contain the action $a$ but solely the latent $z$.

**Algorithm 1** AM Training Procedure
1: Initialize $\pi^{\phi}_o$, and $Q^{\psi}$ (AM-SAC) or $V^{\psi}$ (AM-PPO)
2: Initialize buffer $\mathcal{D} \leftarrow \{\}$
3: Load or pretrain $\pi^{\theta}_f$ using Algorithm 2
4: $s \leftarrow$ env.reset()
5: **for** 1 **to** Interaction Steps **do**
6:     $\mu, \sigma \leftarrow \pi^{\phi}_o(s)$                      ▷ Get policy parameters
7:     $x \sim \mathcal{N}(\cdot|\mu, \sigma)$                    ▷ Sample from Gaussian
8:     $z \leftarrow \tanh(x)$                      ▷ Squash into latent space
9:     $a \leftarrow \pi^{\theta}_f(s, z)$                      ▷ Map to action space
10:    $\hat{V}_s \leftarrow V^{\psi}(s)$                        ▷ for AM-PPO
11:    $\log\pi \leftarrow \log\mathcal{N}(x|\mu, \sigma) - \log(1 - z^2)$     ▷ for AM-PPO
12:    $s', r, d \leftarrow$ env.step($a$)
13:    $\mathcal{D} \leftarrow \mathcal{D} \cup \begin{cases} (s, z, r, s', d), & \text{for AM-SAC} \\ (s, z, \log\pi, \hat{V}_s, r, s', d), & \text{for AM-PPO} \end{cases}$
14:    $s \leftarrow s'$ **if** $\neg d$ **else** env.reset()
15:    **if** $\mathcal{D}$ is ready for training **then**
16:        **if** AM-SAC **then** $\pi^{\phi}_o, Q^{\psi} \leftarrow$ SAC train step
17:        **if** AM-PPO **then** $\pi^{\phi}_o, V^{\psi} \leftarrow$ PPO train epoch, $\mathcal{D} \leftarrow \{\}$

When the buffer is full (AM-PPO) or the buffer has sufficient samples for a batch (AM-SAC), the agent is trained (line 15). For AM-SAC, a standard SAC training step is performed in which the critic is updated to estimate the state-latent value, and the actor maximizes the critic's output with an added entropy term (line 16). For AM-PPO (line 17), the agent is trained on the full buffer as done in PPO, with the critic updated to predict the state value and the actor through standard policy optimization on the latent distribution. In AM-PPO, the buffer is reset after training.

In principle, $\pi^{\theta}_o$ could be trained on the actions $a$ instead of the latent $z$. However, this leads to problems in AM-PPO and AM-SAC. In AM-PPO, it would be challenging to estimate the log-likelihood of an action $a$ given the log-likelihood of latent $z$. While $\pi^{\phi}_f$ is trained to map uniformly into the set of feasible actions, it is neither perfect nor strictly bijective, and the log-likelihood for $a$ would require approximations, e.g., through KDEs. This is costly and likely creates ill-posed gradients for the training of $\pi^{\theta}_o$. In AM-SAC, if training $Q^{\psi}(s, a)$, the policy gradient for $\pi^{\theta}_o$ could propagate through $\pi^{\phi}_f$. While this is tractable, $Q^{\psi}(s, a)$ is not trained on infeasible actions and thus likely yields arbitrary gradients near the feasibility border, preventing $\pi^{\theta}_o$ from jumping between disconnected sets of feasible actions. Additionally, the entropy in the action space is difficult to assess, similar to the log-likelihood. Preliminary experiments on training AM-SAC on $a$, with entropy in the latent space, showed no advantage compared with standard SAC training.

Consequently, training $\pi^{\theta}_o$ and, in AM-SAC, $Q^{\psi}$ on the latent space is significantly more straightforward to implement and yields better results. A primary advantage of training on the latent space is that disconnected sets of feasible actions are very close in the latent space, allowing policy gradient and policy optimization algorithms to jump between the sets. Additionally, a single modal Gaussian in the latent space can be mapped to a multi-modal distribution in the action space, allowing for better exploration and decreasing the chance of being trapped in local optima. Fig. 5 highlights and discusses the exploration benefit. Overall, as shown in Fig. 1, the idea is to convert the SCMDP into

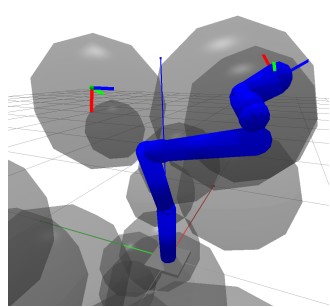

Figure 2: Robotic arm end-effector pose environment with obstacles in gray and the target pose to the left.

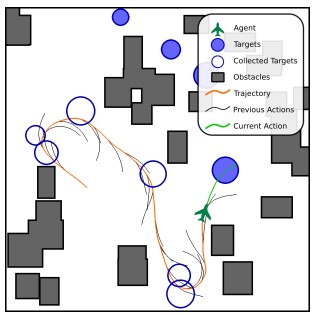

Figure 3: Spline-based path planning environment with constant velocity and non-holonomic constraints.

an unconstrained MDP. Therefore, from the perspective of $\pi_o^\theta$ and $Q^\psi$, the environment is given by $P_f(s'|s, z) = P(s'|s, \pi_f(s, z))$.

## 5 EXPERIMENT SETUP

### 5.1 APPLICATIONS

We define two different RL environments, shown in Figs. 2 and 3, with continuous state and action spaces to demonstrate how action mapping can be implemented and to evaluate its performance compared with common approaches. The first experiment is a robotic arm end-effector pose tracking task with multiple obstacles. This task was designed so that a perfect feasibility model can be derived and a feasible null action exists. The second experiment is a path planning problem with constant velocity and non-holonomic constraints, which can be found using fixed-wing aircraft. Since a fixed-wing aircraft cannot stop or turn around instantaneously, deriving a perfect feasibility model is extremely challenging. Therefore, we utilize that environment to showcase action mapping's performance using approximate feasibility models. In both environments, the episode is terminated when a constraint is violated, which exacerbates the challenge for DRL.

#### 5.1.1 ROBOTIC ARM END-EFFECTOR POSE

In this environment, visualized in Fig. 2, the agent is a purely kinematic robot arm, neglecting inertia, loosely replicating a 7 DOF Franka Research 3 robotic arm (Franka-Robotics, 2024). Given a starting pose, the agent needs to move the joints such that its end-effector reaches a target pose without colliding with obstacles. The obstacles are represented by spheres, and the collision shape of the robot arm is defined by a series of capsules that can be seen as a pessimistic safety hull. The obstacles are sampled using rejection sampling to avoid intersections with the start and end configuration.

**State space.** The state contains the 7 joint angles of the robot arm, the target pose (rotation + translation from the origin), and the parameters of up to 20 spherical obstacles.

**Action space.** The action is defined as a delta of the joint angles.

**Constraints.** The agent is not allowed to exceed its joint limits or predefined maximum cartesian velocities of each joint. No part of the robot arm is allowed to collide with any of the obstacles.

**Reward function.** The reward function is a weighted sum of the decrement of the distance and angle of the end-effector pose to the target pose.

**Feasibility model.** The feasibility model performs a one-step prediction and evaluates the constraint functions on joint limits, joint Euclidean velocities, and obstacle collision.

In this scenario, the feasibility model is perfect, and a feasible replacement action exists (no movement). We train different PPO configurations and compare their performance in the next section.

### 5.1.2 Non-holonomic Path Planning with Constant Velocity

This environment contains an agent that needs to collect targets while avoiding obstacles. The agent can be thought of as a fixed-wing aircraft that needs to maintain a constant velocity, and its turns cannot exceed a maximum curvature. The airplane in Fig. 3 is for visualization purposes only; the agent's dimensions are assumed to be integrated into the obstacles.

**State space.** The state space contains 30 randomly sampled rectangular obstacles and 10 randomly placed and sized circular targets. Additionally, the agent has a position and current velocity.

**Action space.** The agent parameterizes 2D cubic Bezier curves, which are anchored at the agent's position and starting in the agent's current direction, yielding a 5D action space. The splines are followed for a constant time, after which a new spline is generated by the agent.

**Constraints.** The agent is not allowed to collide with any obstacle or leave the squared area. While following the spline for a constant following time, the induced curvature must not exceed a curvature bound, and the agent must not reach the end of the spline.

**Reward function.** The agent receives a reward of 0.1 when a target is collected and an additional reward of 1.0 when all targets are collected.

**Feasibility model.** The approximate feasibility model generates 64 points along the spline and locally assesses collisions with obstacles, whether a point is out of bounds, and whether the local curvature exceeds the curvature bound. Additionally, it adds the Euclidean distances between the points to estimate the length of the spline. The spline length needs to be within length bounds.

The idea behind the spline-based action space is to express a multi-step action with reduced dimensionality. Through this multi-step action, the feasibility model can assess whether a feasible path exists within a time horizon, effectively expressing a short horizon policy that minimizes the future trajectory cost in the second term of equation 7 . Therefore, the minimum length of the generated spline for the feasibility model is set to a multiple of the distance traveled per time step (a factor of 2.5 in this experiment). The maximum length of a spline is defined to bound the action space (3.5 in the experiment), yielding a look-ahead of around two timesteps. We train different SAC configurations and compare their performance in the next section. Our neural network architectures and hyperparameters for these environments are presented in Appendix B. [1]

## 5.2 Comparison

Given a feasibility model G from equation 7 or its Boolean-valued version g from equation 8, the three common approaches to utilize it are action replacement, resampling, and projection. These approaches are described in more detail in Appendix C.1. They each offer distinct trade-offs in terms of computational cost and feasibility guarantees, and our experiments explore their performance in different settings.

Additionally to these methods utilizing feasibility models, we compare with model-free Lagrangian methods "Lagrangian SAC" (Ha et al., 2020) and "Lagrangian PPO" (Ray et al., 2019). In these methods, a safety critic is trained to estimate the expected cumulative cost or, in our case, the expected probability of constraint violation, and a policy aims to maximize a Lagrangian dual problem. The two algorithms are described in more detail in Appendix C.2.

## 6 Results

### 6.1 Robot Arm

To demonstrate the training performance in the robotic arm environment, Figs. 4a and 4b show the cumulative return and constraint violations throughout training. It can be seen that the baseline PPO agent learns robustly, continuously increasing performance, even though showing high failure rates. The Lagrangian PPO shows better constraint satisfaction with similar objective performance. In contrast, action replacement and resampling appear to hamper performance. PPO with action replacement struggles to learn anything in the beginning, presumably because most proposed actions

---

[1]The code and models will be open-sourced after acceptance.

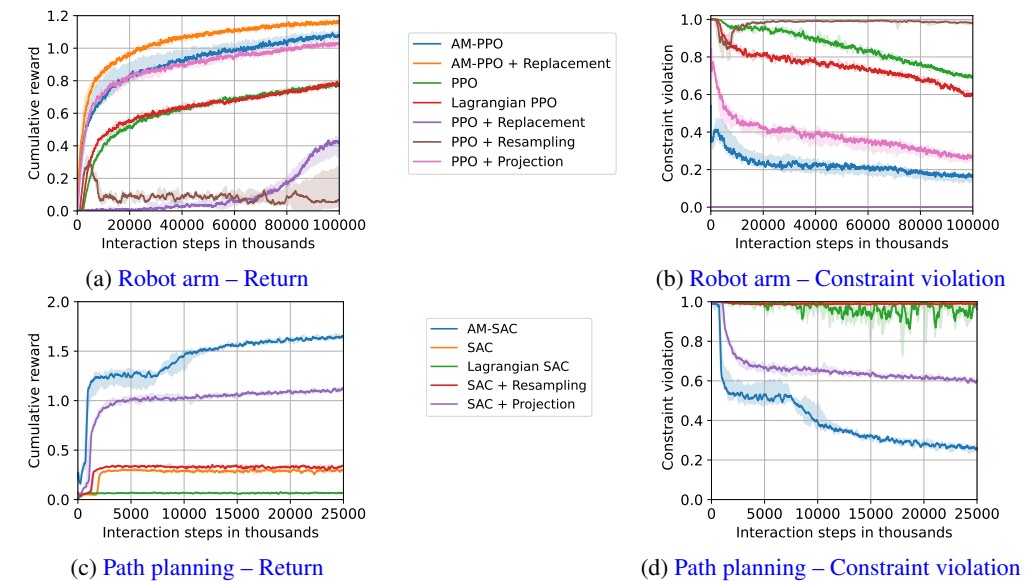

(a) Robot arm – Return

(b) Robot arm – Constraint violation

(c) Path planning – Return

(d) Path planning – Constraint violation

Figure 4: Training curves for the two applications with the results for the robot arm environment in (a)+(b) and for the path planning environment in (c)+(d). For each configuration, 3 agents were trained, with the curves showing the median and the region between highest and lowest performance. Both "Replacement" agents show no constraint violation in (b).

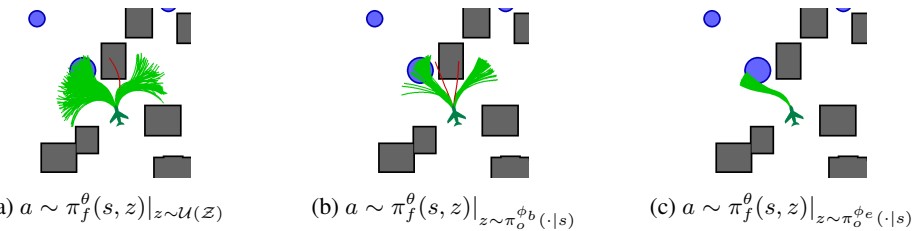

(a) $a \sim \pi_f^\theta(s,z)|_{z \sim \mathcal{U}(\mathcal{Z})}$     (b) $a \sim \pi_f^\theta(s,z)|_{z \sim \pi_o^{\phi_b}(\cdot|s)}$     (c) $a \sim \pi_f^\theta(s,z)|_{z \sim \pi_o^{\phi_e}(\cdot|s)}$

Figure 5: Visualization of 256 generated actions of $\pi_f^\theta$ for a given state (a) if $z$ is sampled uniformly, (b) if sampled from the distribution of $\pi_o^{\phi_b}$, being an objective policy in the beginning of training, and (c) if sampled from the distribution of $\pi_o^{\phi_e}$ which is the agent at the end of training.

are replaced with the null action. Therefore, its failure rate is constant at zero. PPO with action re-sampling first learns faster than PPO but then exhibits instability, likely due to the wrong estimation of the policy ratio in the PPO objective.

PPO with action projection and action mapping (AM-PPO, AM-PPO + Replacement) learn significantly faster with higher final performance than the model-free baselines. Action mapping yields slightly higher performance at the end of training, and it exhibits fewer constraint violations than action projection. Adding action replacement to action mapping yields the best performance without constraint violations. This application showed that action mapping and projection are both very beneficial with perfect feasibility models that are mostly convex, with action mapping having a slight edge. Since in this example a safe action exists, action replacement can always be added to guarantee constraint satisfaction and together with action mapping, it also improves performance. Additionally, an evaluation of training and inference times in Appendix F shows that projection is significantly more expensive than action mapping.

## 6.2 PATH PLANNING

Before inspecting the training performance of the different approaches, Fig. 5 shows the inner workings of action mapping. Fig. 5a shows the output distribution of the pretrained feasibility policy $\pi_f^\theta$ when sampling uniformly from the latent space $\mathcal{Z}$. The feasibility policy is able to generate actions in both disconnected sets of feasible actions, with only minimal actions between. Figs. 5b and 5c

show how the objective policy can take advantage of this. At the beginning of training, the agent outputs a distribution with high entropy in the latent space, leading to a bi-modal distribution in the action space (Fig. 5b). After training, the agent's output entropy is lower, and the resulting distribution in the action space usually collapses to a single mode (Fig. 5c). This visualization shows a crucial aspect of action mapping. It allows an objective policy to express multi-modal action distributions by only parameterizing a single Gaussian distribution in the latent space, which can significantly improve exploration.

Inspecting the training performance of the different approaches in the path planning environment in Figs. 4c and 4d, it can be seen that SAC and SAC with resampling do not learn much. Their initial jump in performance occurs when the agent starts understanding the targets, but it usually fails to reach them because it does not learn to understand the obstacles. Therefore, both agents always end their episodes through a constraint violation. The Lagrangian SAC agent focuses only on not violating constraints, which leads to slightly better constraint satisfaction but completely inhibits the learning of the objective. In contrast, when the action projection agent learns to understand the targets, its performance jumps significantly higher. The reason is that when it tries to go straight to each target, action projection pushes it around the obstacles.

The action mapping agent (AM-SAC) exhibits a higher initial performance jump, indicating that action mapping more successfully nudges the agent around obstacles. Additionally, in contrast to all other approaches, the action mapping agent has a second jump in performance and constraint satisfaction between 5M and 10M steps. This leap can be attributed to the objective policy's understanding of the obstacles and incorporating them into the plan instead of only being nudged around by the feasibility policy. This application shows that action mapping outperforms action projection with approximate feasibility models. Fig. A.2 in the appendix shows trajectory examples of the AM-SAC agent, highlighting the difficulty of that environment through the high variability of initial conditions. An evaluation of the dependence on the approximation accuracy of the feasibility model is shown in Appendix G. Additionally, as in the robotic arm example, training and inference times of action mapping are significantly faster than action projection, as shown in Appendix F.

# 7 CONCLUSION AND FUTURE WORK

We proposed and implemented a novel DRL training strategy based on action mapping. Our results demonstrate that this approach performs exceptionally well, particularly when using approximate feasibility models. We highlighted how even approximate model knowledge can be effectively incorporated into the DRL process to enhance training performance, emphasizing the potential to integrate domain-specific insights into DRL frameworks. We further show how action mapping allows the agent to express multi-modal action distributions, which can significantly improve exploration.

The use of KDE introduces some distance between the generated actions and the boundary of feasible actions, which may result in conservative action selection. Furthermore, the feasibility policy $\pi_f$ does not completely eliminate the generation of infeasible actions and, therefore, does not provide strict safety guarantees. Consequently, the learned $\pi_f^\theta$ is not surjective and does not remove all constraints from the SCMDP, but still significantly relaxes the constraints.

While the assumption of having a feasibility model may be too restrictive in general, we show that deriving one and utilizing action mapping can substantially improve learning performance. Therefore, we advocate for exploring the utilization of feasibility models in practical applications where model-free RL's performance is insufficient.

Future work will explore whether weight-sharing or initialization of parameters from $\pi_f$ to the actor and critic of the policy $\pi_o$ could lead to more efficient learning. Furthermore, spline-based path planning, which has shown promising results, warrants further investigation, particularly in robotic path planning scenarios. Lastly, expanding action mapping to utilize learned feasibility models could be explored to offer an alternative to common approaches like Lagrangian multipliers or action projection.

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

APPENDIX

# A  FEASIBILITY POLICY TRAINING

---

**Algorithm 2** Feasibility Policy Pretraining, adapted from Theile et al. (2024)

---

1: Initialize $\pi_f^\theta$
2: **for** 1 **to** Feasibility Training Steps **do**
3:     **for** $k = 1$ **to** $K$ **do**
4:         $s_f \leftarrow$ Generate partial state in $\mathcal{S}_f$         ▷ Only containing feasibility relevant information
5:         $z_i \sim \mathcal{U}(\mathcal{Z}), \quad \forall i \in [1, N]$                          ▷ Sample uniformly in latent space
6:         $a_i \leftarrow \pi_f^\theta(s_k, z_i), \quad \forall i \in [1, N]$                ▷ Map latents to actions for given state
7:         $a_j^* \leftarrow a_j + \epsilon_j, \quad \epsilon_j \sim \mathcal{N}(0, \sigma'), \quad \forall j \in [1, N]$     ▷ Add noise to actions to get samples
8:         $\hat{q}_j \leftarrow \frac{1}{N} \sum_{i=1}^{N} k_\sigma(a_j^* - a_i), \quad \forall j \in [1, N]$               ▷ Evaluate KDE on samples
9:         $\hat{q}_j' \leftarrow \frac{1}{N} \sum_{i=1}^{N} k_{\sigma'}(a_j^* - a_i), \quad \forall j \in [1, N]$          ▷ Evaluate proposal KDE on samples
10:         $r_j \leftarrow g(s_k, a_j^*), \quad \forall j \in [1, N]$                 ▷ Evaluate feasibility model on samples
11:         $\hat{Z}_k \leftarrow \frac{1}{N} \sum_{j=1}^{N} \frac{r_j}{\hat{q}_j'}$                                    ▷ Estimate partition function
12:         $\hat{p}_j \leftarrow \frac{r_j}{\hat{Z}_k}, \quad \forall j \in [1, N]$                       ▷ Estimate target distribution at the samples
13:         $g_k \leftarrow \frac{1}{2N} \sum_{j=1}^{N} \frac{\hat{q}_j}{\hat{q}_j'} \log\left(\frac{2\hat{q}_j}{\hat{q}_j + \hat{p}_j}\right) \nabla_\theta \log(\hat{q}_j)$           ▷ Compute the gradient of the JS loss
14:     $\theta \leftarrow \theta - \alpha_\theta \frac{1}{K} \sum_{k=1}^{K} g_k$                                     ▷ Apply gradient

---

To determine how long the feasibility policy needs to be trained, the policy should be evaluated at regular intervals. This is necessary since the feasibility model is only evaluated on noisy samples (line 10), and thus, the result cannot be used to determine convergence. If the agent's precision (i.e., the number of feasible actions among all generated actions) and the average distance between feasible actions (an indicator of recall, i.e., higher distance means covering more feasible actions) stabilizes, the agent has trained sufficiently. Note that the precision usually cannot and does not need to reach 100%, as the agent will always generate infeasible actions if the sets of feasible actions are disconnected. Additionally, the partial state generator may sometimes generate states for which no feasible action exists, as discussed in the following.

## A.1  PARTIAL STATE GENERATOR

For the feasibility policy training, a partial state generator and a parallelizable feasibility model $g : \mathcal{S}_f \times \mathcal{A} \to \mathbb{B}$ are needed. In most environments, not all state variables are relevant for feasibility and can thus be omitted in the feasibility policy training (e.g., the end-effector pose target in the robot arm environment and the target regions in the path planning environment). Removing these variables from the state space yields the partial state space $\mathcal{S}_f$.

The state generator does not need to generate states with a distribution similar to a realistic state visitation from any policy. It is even preferential to generate more safety-critical partial states. Therefore, more obstacles can be generated, and the agent's position can be sampled closer to these obstacles than when generating initial states for episodes.

In the robotic arm environment, the state generator generates randomly placed spherical obstacles and a random joint configuration within the joint limits. The partial state is ready after removing obstacles that collide with the arm.

In the path planning environment, the state generator similarly generates randomly placed rectangular obstacles and a random position and velocity of the agent. After removing obstacles colliding with the position, the partial state is ready. Our experiments show that ensuring a feasible action exists for every generated state is unnecessary.

## B   NEURAL NETWORK ARCHITECTURE AND HYPERPARAMETERS

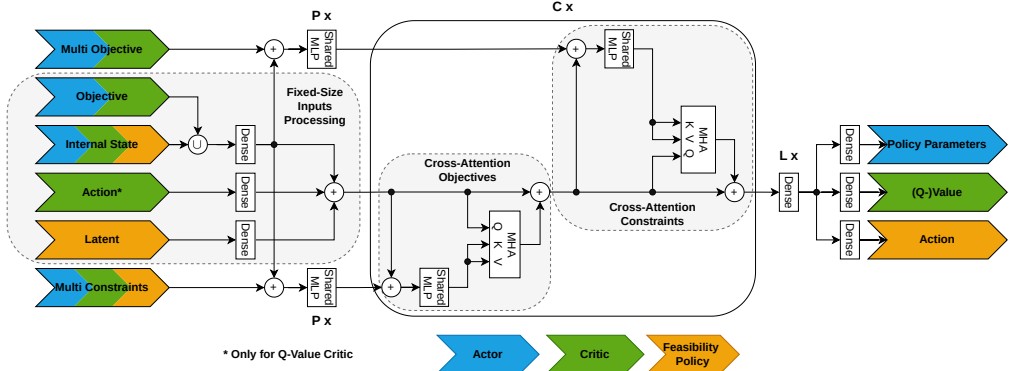

Figure A.1: Generalized neural network architecture for the networks used in both experiments. The different network types ($\pi_o^\theta$, $Q^\psi$, $V^\psi$, $\pi_f^\phi$) have different inputs and outputs indicated through the coloring. As such, the feasibility policy does not process objective-related information, omitting the second cross-attention. While all networks share the same structure, they do not share parameters. The Shared MLPs before the cross-attention are repeated $P$ times, the cross-attention layers are repeated $C$ times, and the dense layer after them is repeated $L$ times. The $\cup$ is a concatenation. Layer-Normalization is added for all inputs to the Multi-Head Attention (MHA). The hyperparameters are listed in Table A.1.

Both environments contain lists of elements, such as the obstacles in the robotic arm environment and the obstacles and targets in the path planning one. The agent needs to process these lists invariantly to permutations. Therefore, we utilize attention, with its query given by the internal state representation of the agent, and key and value being the obstacles and targets alternatingly. All networks, the actor, critic, and feasibility policy share the same architecture, with the exception that the feasibility policy does not process the objective-relevant inputs.

## C   COMPARISON BASELINES

### C.1   FEASIBILITY MODEL-BASED

**Action Replacement.** The feasibility model g is queried on a proposed action of the agent. If deemed feasible, the action is applied to the system. Otherwise, the action is rejected and replaced with a predefined feasible one. The feasible action is usually task-independent and does not contribute meaningfully to task completion. Additionally, it is often not trivial to derive a feasible action, such as in the path planning environment. Therefore, we only compare to action replacement in the robotic arm example, where the feasible action is to apply no motion to the arm.

**Action Resampling.** Similar to action replacement, infeasible actions are rejected. Instead of replacing them with a known feasible action, the stochastic agent is queried again to provide different actions. This resampling step is repeated until either a feasible action is found or a maximum number of sampling steps is reached. The advantage is that it does not require a known feasible action. However, if the agent's action distribution is too far from the feasible actions, resampling may fail to generate a feasible action within the allowed iterations, leading to high failure rates or inefficient learning. It further alters the actual likelihood of actions, which we show can destabilize the DRL training.

**Action Projection.** In this approach, an optimization method finds the closest feasible action to the one proposed by the agent based on some distance metric. While action projection often yields better task performance by staying close to the agent's intended action, it can introduce computational overhead since the optimization process must be repeated at every step. For example, Donti et al. (2021) propose iteratively minimizing G from equation 7 via gradient steps $\Delta_a G(s, a)$. However,

Table A.1: Hyperparameters for the AM-SAC and AM-PPO configurations.

| Parameter | AM-SAC | AM-PPO |
|---|---|---|
| Total interaction steps | $25,000,000$ | $100,000,000$ |
| Memory/Rollout size | $1,000,000$ | $10,000$ |
| Batch size | 128 | 128 |
| Discount factor ($\gamma$) | 0.97 | 0.97 |
| Actor learning rate (initial) | $3 \times 10^{-5}$ | $3 \times 10^{-5}$ |
| Actor learning rate decay rate | - | 0.0 |
| Actor learning rate decay steps | - | $100,000,000$ |
| Critic learning rate (initial) | $1 \times 10^{-4}$ | $1 \times 10^{-4}$ |
| Critic learning rate decay rate | - | 0.0 |
| Critic learning rate decay steps | - | $100,000,000$ |
| Entropy coefficient | 0.0002 | 0.005 |
| Soft update factor ($\tau$) | 0.005 | - |
| Policy update delay | 2048 | - |
| Training steps per environment step | 2/50 | - |
| Number of parallel environments | 50 | 50 |
| Number of rollout epochs | - | 3 |
| Advantage normalization | - | true |
| GAE lambda ($\lambda$) | - | 0.9 |
| Clipping parameter ($\epsilon$) | - | 0.2 |
| Feasibility divergence samples ($N$) | 1024 | 1024 |
| Feasibility divergence sigma ($\sigma$) | 0.1 | 0.1 |
| Feasibility divergence sigma prime factor | 2.0 | 1.0 |
| Feasibility Training Steps | $500,000$ | $1,000,000$ |
| Feasibility states per batch ($K$) | 16 | 16 |
| Feasibility learning rate | $1 \times 10^{-4}$ | $1 \times 10^{-4}$ |
| Layer Size | 256 | 256 |
| Num Dense Layers Pre Attention ($P$) | 1 | 3 |
| Num Cross-Attention Layers ($C$) | 3 | 3 |
| Num Heads in MHA | 16 | 4 |
| Key Dim in MHA | 16 | 64 |
| Num Dense Layers Post Attention ($L$) | 3 | 3 |

if G is not convex–which is often the case in practical scenarios–action projection may not yield a feasible solution, as the optimization process can get trapped in local minima. Furthermore, the actions are projected onto the boundary between feasible and infeasible actions. If the feasibility model is only approximate, the actions on the boundary may not be feasible, requiring more conservative approximate solutions. Specifically, in the path planning environment, a higher distance to obstacles and a tighter curvature bound had to be enforced.

## C.2 MODEL-FREE LAGRANGIAN ALGORITHMS

To use Lagrangian algorithms, a safety critic $Q_C(s, a)$ is trained that estimates the expected cumulative cost or probability of failure according to

$$Q_C(s, a) = c(s, a) + \gamma_C \mathbb{E}_{s' \sim P(\cdot|s,a)} \left[ \min_{a' \in \mathcal{A}} Q_C(s', a') \right]. \qquad (17)$$

For on-policy algorithms, the corresponding state cost-value is defined as

$$V_C^\pi(s) = \mathbb{E}_{a \sim \pi(\cdot|s)} \left[ Q_C(s, a) \right]. \qquad (18)$$

For the Lagrangian SAC (Ha et al., 2020), the objective of the policy is the Lagrangian dual

$$\min_{\lambda \geq 0} \max_\pi \mathbb{E}_{s \sim \mathcal{D}, a \sim \pi(\cdot|s)} \left[ Q(s, a) + \alpha \log \pi(a|s) - \lambda(Q_C(s, a) - \delta_C) \right], \qquad (19)$$

with $\delta_C$ being a safety threshold that can be tuned.

For the Lagrangian PPO (Ray et al., 2019), (based on the implementation by OmniSafe(OmniSafe Team, 2022)), the policy objective is the dual

$$\min_{\lambda \geq 0} \max_\pi \mathbb{E}_{(s,a) \sim \pi} \left[ \frac{\pi(a|s)}{\pi_{\text{old}}(a|s)} (A^\pi(s, a) - \lambda A_C^\pi(s, a)) \right], \qquad (20)$$

in which the advantages A and $A_C$ are estimated using the generalized advantage estimation Schulman et al. (2015). As in equation 4, the change in the ratio can be clipped with parameter $\epsilon$. The hyperparameters used are listed in Tab. A.2.

Table A.2: Hyperparameters for the Lagrangian SAC and Lagrangian PPO configurations that are different from Tab. A.1.

| Parameter | AM-SAC | AM-PPO |
|---|---|---|
| Discount factor cost $\gamma_C$ | 0.9 | 0 (Not needed) |
| Safety delta $\delta_C$ | 0.05 | - |
| Safety critic learning rate | $1 \times 10^{-4}$ | $1 \times 10^{-4}$ |
| Lagrangian mult. learning rate | 0.01 | 0.01 |

## D    ROBOTIC ARM ENVIRONMENT

Table A.3: Parameters of the robotic arm environment.

| Description | Value |
|---|---|
| number of steps until *timout* flag is set | 100 |
| time duration of one timestep | 0.5s |
| maximum number of obstacles after rejection sampling | 20 |
| number of obstacles before rejection sampling | 30 |
| maximum allowed cartesian speed of any joint | 0.3 m/s |
| maximum angle change in one timestep | 90° |

Table A.4: DH parameters of the robotic arm (Franka-Robotics, 2024).

| Joint | a [m] | d [m] | $\alpha$ [rad] | $\theta$ [rad] |
|---|---|---|---|---|
| Joint 1 | 0 | 0.333 | 0 | $\theta_1$ |
| Joint 2 | 0 | 0 | $-\frac{\pi}{2}$ | $\theta_2$ |
| Joint 3 | 0 | 0.316 | $\frac{\pi}{2}$ | $\theta_3$ |
| Joint 4 | 0.0825 | 0 | $\frac{\pi}{2}$ | $\theta_4$ |
| Joint 5 | -0.0825 | 0.384 | $-\frac{\pi}{2}$ | $\theta_5$ |
| Joint 6 | 0 | 0 | $\frac{\pi}{2}$ | $\theta_6$ |
| Joint 7 | 0.088 | 0 | $\frac{\pi}{2}$ | $\theta_7$ |
| Flange | 0 | 0.107 | 0 | 0 |

Table A.5: Robotic arm joint limits (Franka-Robotics, 2024).

| Joint | Lower Limit [rad] | Upper Limit [rad] |
|---|---|---|
| Joint 1 | -2.7437 | 2.7437 |
| Joint 2 | -1.7837 | 1.7837 |
| Joint 3 | -2.9007 | 2.9007 |
| Joint 4 | -3.0421 | -0.1518 |
| Joint 5 | -2.8065 | 2.8065 |
| Joint 6 | 0.5445 | 4.5169 |
| Joint 7 | -3.0159 | 3.0159 |

# E PATH PLANNING SOLVED SCENARIOS

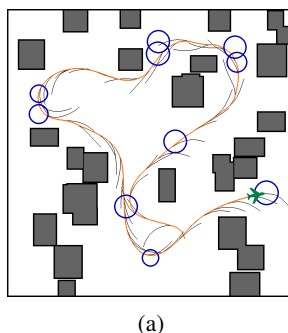 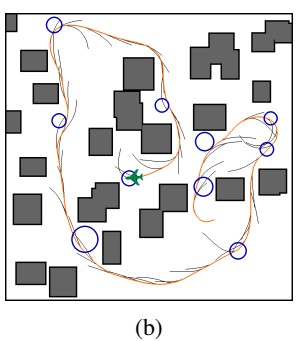 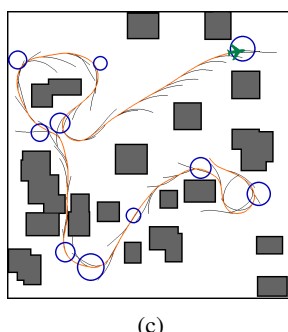

|          (a)          |          (b)          |          (c)          |

Figure A.2: Example trajectories of the AM-SAC agent solving random scenarios of the spline-based path planning environment with non-holonomic constraints.

# F COMPUTATION COMPARISON

Table A.6: Timing measurements average for the different agent configurations in both environments using the machine with specifications in Tab. A.7. Inference time mixed refers to deploying neural network inference on the GPU while keeping projection on the CPU.

|                          | **Robotic Arm** | | | **Path Planning** | | |
|--------------------------|------|------------|------------------|------|------------|------------------|
|                          | PPO  | PPO + Proj. | AM-PPO          | SAC  | SAC + Proj. | AM-SAC          |
| Training time [h]        | 11.6 | 23.5       | 16.8 (5.5 + 11.3) | 13.0 | 22.0       | 16.3 (3.3 + 13.0) |
| Inference time CPU [ms]  | 3.25 | 8.23       | 5.18            | 5.63 | 11.96      | 7.66            |
| Inference time GPU [ms]  | 0.87 | 7.17       | 1.45            | 1.33 | 10.91      | 2.28            |
| Inference time Mixed [ms]| -    | 6.70       | -               | -    | 8.06       | -               |

Tab. A.6 presents a comparison of training and inference times across different agent configurations in the robotic arm and path planning environments. The results are measured on the system described in Tab. A.7.

In terms of training time, both environments exhibit similar trends. After pretraining the feasibility policy, the training time for the policy $\pi_o$ in the action mapping (AM) approaches with SAC and PPO is comparable to their respective baselines. However, the projection-based methods take significantly longer to train, as the projection step is computationally expensive and must be executed sequentially, which adds substantial overhead.

Inference time results further highlight the efficiency of the proposed AM method. Since AM only introduces an additional neural network inference, and the network is relatively smaller (as it does not process the objective-relevant inputs such as target pose or list of targets), the increase in inference time is modest, approximately 50%. On the other hand, projection-based methods incur a much larger time overhead due to the sequential nature of the projection process, which limits the ability to fully utilize GPU acceleration. When comparing mixed projection (GPU for network inference, CPU for projection) with GPU-based AM, the projection approach is approximately three times slower.

These results demonstrate the computational advantages of AM, particularly in scenarios where inference efficiency is crucial.

Table A.7: Computer Specifications

| Specification | Details |
| --- | --- |
| CPU | AMD Ryzen Threadripper PRO 7985WX (64 Cores, 5.1 GHz Boost) |
| RAM | 512 GB DDR5 (4800 MT/s) |
| GPU | NVIDIA GeForce RTX 4090 (24 GB VRAM) |
| Operating System | Ubuntu 22.04.4 LTS |

# G FEASIBILITY MODEL APPROXIMATION

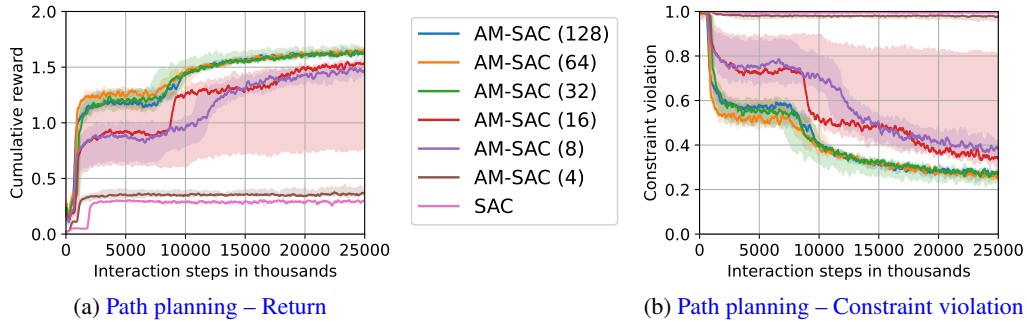

(a) Path planning – Return    (b) Path planning – Constraint violation

Figure A.3: Training curves of different AM-SAC in the path planning environment with different numbers of feasibility evaluation points. For each configuration, 3 agents were trained.

The action space in the path planning environment utilizes splines to parameterize possible multi-step trajectories for which a cost as in equation 6 can be approximated. The cost is approximated in two ways. First, the spline length is bounded since it would require too many parameters to describe the entire trajectory. Therefore, the feasibility model can only determine if a feasible trajectory exists for the next few steps.

The second approximation in the feasibility model is a numerical approximation of the violation of constraints along the spline. To facilitate the fast evaluation of the feasibility model, $S$ equidistant points in parameter space are evaluated for constraint violation (outside environment, inside obstacle, local curvature exceeding maximum). Additionally, the length of the spline is approximated as the sum of Euclidean distances between these points. Therefore, the number of points $S$ plays a significant role in the accuracy of the feasibility model.

To assess the impact of the accuracy of the feasibility model on the learning progress of action mapping, we trained action mapping agents ($\pi_f$ and $\pi_o$) using different numbers of $S \in \{4, 8, 16, 32, 64, 128\}$, in which $S = 64$ corresponds to the results in Section 6.2. The results in Fig. A.3 show that higher values of $S$ lead to better and more stable performance. With $S = 4$, AM-SAC performs only slightly better than SAC, but with more points, the performance jumps significantly. From $S = 32$ and above, the performance does not change significantly anymore. The computational cost of training $\pi_f$ is not very sensitive to $S$ since the points can be evaluated in parallel. The training time of $\pi_f$ for $S = 4$ is approximately 20% faster than for $S = 128$.

