# OpenReview forum: "Action Mapping for Reinforcement Learning in Continuous Environments with Constraints"
_ICLR.cc/2025/Conference — Submitted to ICLR 2025_

### Official Review · Reviewer_YKMm · 2024-10-29

**Soundness:** 2
**Presentation:** 2
**Contribution:** 1
**Rating:** 3
**Confidence:** 4

**Summary:**

This paper proposes a strategy called action mapping for RL in continuous environments with state-wise constraints. The idea is to learn a latent action space and an action mapping model, with which the policy samples a latent action and the latent action is further mapped to a feasible action. The proposed method is evaluated in a robotic arm end-effector pose positioning task and a path planning environment, showing better performance than several existing methods in terms of higher returns and lower constraint violations.

**Strengths:**

- The target problem to solve in this work is of great significance to many real-world applications.
- The related works are introduced and discussed satisfactorily.

**Weaknesses:**

- Although the authors mentioned the reference [Theile et al., 2024], I found the content in Section 4 overlaps largely with the content in Section 3 and Section 4 of [Theile et al., 2024]. For example, Equation 16 in this paper is almost the same as the JS loss in Table 2 of [Theile et al., 2024]. Therefore, the novelty and contribution of this paper are questionable.
- The whole training process is not clear. Adding a pseudocode of the proposed algorithm will help. Especially, the training of the feasibility model is not clear enough. In addition, many technical details are missing. Please see my questions below.
- The idea of action mapping is closely related to the research on action representation learning [1-4]. The illustration in Figure 1 is very similar to the concept presented by Figure 1 in [1]. These related works should be included in the related work section for a detailed discussion.

### Minors

- The symbol $J$ in Equation 1,2 and Equation 3,4 are inconsistent.
- The legends in Figure 4 are too small.

---
### Reference

[1] Yash Chandak, Georgios Theocharous, James E. Kostas, Scott M. Jordan, Philip S. Thomas. Learning Action Representations for Reinforcement Learning. ICML 2019: 941-950

[2] Boyan Li, Hongyao Tang, Yan Zheng, Jianye Hao, Pengyi Li, Zhen Wang, Zhaopeng Meng, Li Wang. HyAR: Addressing Discrete-Continuous Action Reinforcement Learning via Hybrid Action Representation. ICLR 2022

[3] Pengjie Gu, Mengchen Zhao, Chen Chen, Dong Li, Jianye Hao, Bo An. Learning Pseudometric-based Action Representations for Offline Reinforcement Learning. ICML 2022: 7902-7918

[4] William F. Whitney, Rajat Agarwal, Kyunghyun Cho, Abhinav Gupta. Dynamics-Aware Embeddings. ICLR 2020

**Questions:**

1. For the training in Section 4.1, does it require a ground-truth $g(s,a)$?
2. What is the bound of the output of $\pi_f$ and $\pi_o$?
3. What are the implementation details of the proposed method, e.g., network structure, hyperparameters?
4. For Figure 4, why is SAC+Replacement not included in Figure 4c? And where is the constraint violation plot for SAC?
5. Throughout the paper, it seems that the definitions in Equation 5-7 are not necessary, as in Section 4 and the experiments only the function $g$ is assumed. Can the authors provide more explanation on this point?
6. How many seeds/trials are used in Figure 4?

---

> ### Author Response · Authors · 2024-11-28
> **Answers to Questions**
>
> Thank you very much for the detailed feedback and literature recommendations. We adapted the manuscript accordingly and hope that we were able to address all concerns.
>
> ### To Weakness 1:
> You are right. We did not clearly discuss the difference with Theile et al. (2024). The revised manuscript now discusses the difference clearly and emphasizes the contribution of this work. To summarize, Theile et al. (2024) only derived how to train a feasibility policy given a feasibility model and hypothesized that it could benefit DRL through an action mapping framework. However, they did not formulate a way to train the objective policy and did not test their hypothesis in DRL. In this work, we describe how to actually use the action mapping framework in DRL.
>
> ### To Weakness 2:
> We added Algorithms 1+2 and a description that should clarify the training process. Additionally, we added the requested technical details.
>
> ### To Weakness 3:
> Thank you for pointing us in that direction. We added a description to the related work section. Action mapping can be thought of as learning an action representation for the state-dependent set of feasible actions, which we incorporated throughout the manuscript.
>
> ### To Minor Weaknesses:
> Thank you, we adapted the manuscript accordingly.
>
> ### Question 1:
> We do require a g(s, a), but it does not need to be perfect or ground-truth. In the robotic arm environment, it was perfect, but in the path planning environment, it was only an approximation. We elaborate further on the approximation in Appendix G.
>
> ### Question 2:
> $\pi_f$’s output is the action space, which we commonly bound between -1 and 1 for all action dimensions and then scale it to the real action bounds within the environment. The output of $\pi_o$ are parameters of a Gaussian that is then squashed with a tanh to produce latent values in the latent space, which is also bound between -1 and 1. We added Algorithm 1, which further elaborates on this point.
>
> ### Question 3:
> We added the network structure and hyperparameters in Appendix B. Also, Algorithms 1+2 provide further implementation details.
>
> ### Question 4:
> Replacement can only be added if a feasible action is known. However, deriving a feasible action is often significantly harder than assessing whether a proposed action is feasible given a state. In the robotic arm, a default feasible action is the null action (no movement), which can be used as the replacement action. In the path planning environment, a null action does not exist since the airplane needs to continue flying. Therefore, we do not have a feasible replacement action in the second environment and cannot train SAC + replacement.
>
> We added the constraint violation plot for SAC.
>
> ### Question 5:
> We elaborated a bit more on what the spline-based action space is achieving:
> “The idea behind the spline-based action space is to express a multi-step action with reduced dimensionality. Through this multi-step action, the feasibility model can assess whether a feasible path exists within a time horizon, effectively expressing a short horizon policy that minimizes the future trajectory cost in the second term of equation 7.“
> Our intent of equation (7) is to show the components of a feasibility model, which is the immediate cost (equation (5)) plus the best expected future cost (equation (6)).
>
> ### Question 6:
> For each configuration, we trained on 3 seeds, as indicated by the caption of Fig. 4.

---

### Official Review · Reviewer_vBAY · 2024-11-01

**Soundness:** 1
**Presentation:** 1
**Contribution:** 2
**Rating:** 3
**Confidence:** 4

**Summary:**

This paper proposes an action mapping method that distinguishes between feasibility and objective policies during training. By pretraining the feasibility policy first and then training the objective policy, the approach enables more efficient learning within a reduced action set. Experimental results demonstrate that the proposed method results in fewer constraint violations and achieves higher returns compared to previous action replacement, resampling, and projection methods.

**Strengths:**

1. The method for training the feasibility policy is novel, allowing for fewer constraint violations and higher returns compared to other methods.
2. Experiments were conducted in environments requiring constraints, such as a robotic arm task and a spline-based path planning.
3. The approach is straightforward and can be combined with any RL algorithm.

**Weaknesses:**

1. The assumption that the feasible policy can be pretrained seems overly strict. Pretraining requires prior knowledge of the cost function $C^\tau(s;\pi)$ and the feasibility model $G(s,a)$, which may be difficult to assume in general.
2. The experimental environments appear limited. It would be beneficial to include comparisons in environments like Safety Gym or other constrained RL environments.
3. There is a lack of baseline algorithms. Currently, the comparisons are limited to variants of action mapping, such as action resampling and projection. Direct comparisons with a wider range of methods, including Lagrangian approaches, would strengthen the evaluation. Even if these methods are primarily designed for standard CMDPs rather than SCMDPs, adjusting the constraint thresholds to be more strict would allow for a fair comparison under the SCMDP constraints used in the original experiments.

**Questions:**

1. **[About Weakness 1]** Isn’t the required setup for the proposed method too strict? Other constrained RL methods estimate the cost function without prior knowledge, obtaining cost information similarly to rewards. In such cases, would the proposed action mapping approach still be applicable?

2. **[About Weakness 2]** Could you provide experimental results in a wider variety of environments?

3. **[About Weakness 3]** Could you also show performance comparisons with other constrained RL methods?

---

> ### Author Response · Authors · 2024-11-28
> **Answers to Questions**
>
> Thank you very much for your feedback. We updated the manuscript accordingly.
>
> ### Question 1:
> Being able to derive a feasibility model may indeed not be applicable for all environments. However, we show how much performance improvement can be gained if one is derived and action mapping is utilized; thus, it might be worthwhile to try deriving one if model-free RL is insufficient. We added a paragraph in the Conclusion that elaborates on this point.
>
> In principle, a safety critic could be trained on offline trajectory data to assess the cumulative cost of actions. The feasibility policy could then be pretrained to generate all actions for which the expected cumulative cost is less than some threshold. This approach can be interesting to test in future work, but it is also challenging since it requires large amounts of trajectory data that includes sufficient numbers of constraint violations.
>
> ### Question 2:
> Unfortunately, we were not able to add more environments during this rebuttal phase. However, we added further experiments for the given environments and hope that those can better support the benefits of action mapping.
>
> ### Question 3:
> We originally thought a comparison with model-free approaches is unfair since they do not have access to the inductive bias of the feasibility model. However, we added a Lagrangian PPO and SAC implementation to the comparison. They do improve constraint satisfaction slightly compared to PPO and SAC but do not help or even hamper performance. Overall, action mapping and action projection perform significantly better.
>
> When searching for the best parameters, specifically for Lagrangian SAC, we noticed that tweaking them only yields two outcomes. Either they are strict, yielding slightly improved constraint satisfaction but hampering performance compared to SAC, or loose, yielding no improvement in constraint satisfaction and similar objective performance. In the paper, we show the stricter version since it at least improves constraint satisfaction.

---

### Official Review · Reviewer_yMya · 2024-11-04

**Soundness:** 3
**Presentation:** 3
**Contribution:** 3
**Rating:** 6
**Confidence:** 4

**Summary:**

This paper tackles the problem of how to efficiently train agents in environments with constraints (like a robotic arm avoiding obstacles or an aircraft maintaining non-holonomic constraints). Traditional DRL approaches struggle with poor sample efficiency and slow convergence in such constrained environments. The authors propose "action mapping," which decouples the learning process into two parts: first training a feasibility policy that learns to generate all feasible actions for any state, then training an objective policy to select optimal actions from this feasible set, effectively transforming a state-wise constrained Markov Decision Process (SCMDP) into an unconstrained MDP. They validate their approach through two experiments - a robotic arm end-effector pose task with perfect feasibility models and a path planning problem with approximate feasibility models - demonstrating superior performance compared to common approaches like action replacement, resampling, and projection.

**Strengths:**

- The most significant contribution is how action mapping allows agents to express multi-modal action distributions through a simple Gaussian in latent space, improving exploration. Ability to plan with approximate feasibility models is a notable advantage since perfect models are rarely available in practical applications.

**Weaknesses:**

- The paper omits constraint violation plots for the path planning task, making it impossible to verify claims about performance with approximate feasibility models.

- The feasibility model is a critical component of the proposed architecture, yet the paper lacks essential analysis and ablations of this component. Key questions remain unanswered: How is state space sampling performed during training? What metrics determine sufficient training of the feasibility model? How does the quality/approximation level of the feasibility model impact overall performance? Without these analyses, it's difficult to understand the method's robustness and its applicability to scenarios where perfect or near-perfect feasibility models aren't available.

Minor Comments:

- Typo on line 406, “actiosn”

**Questions:**

NA

---

> ### Author Response · Authors · 2024-11-28
> **Addressing Weaknesses**
>
> Thank you very much for the feedback. We incorporated all of your comments in the revised manuscript:
>
> ### Weakness 1:
> We added the constraint violation plot for the path planning task in Fig. 4(d), which shows the same trend as the return plot in Fig. 4(c).
>
> ### Weakness 2:
> We added a description of the feasibility policy algorithm, state space sampling, and metrics for sufficient training in Appendix A. Additionally, we added an ablation on the accuracy of the feasibility model in Appendix G. We reduced and increased the points along the spline for which feasibility was assessed. It can be seen that higher accuracy leads to better performance, as was expected.
>
> We fixed the actiosn typo.

---

### Official Review · Reviewer_8UnY · 2024-11-04

**Soundness:** 2
**Presentation:** 2
**Contribution:** 3
**Rating:** 5
**Confidence:** 3

**Summary:**

The authors introduce a novel approach known as action mapping (AM) within the context of deep reinforcement learning (DRL), showcasing its effectiveness, particularly when utilizing approximate feasibility models. Their results suggest that the integration of approximate model knowledge can improve training performance and enable agents to represent multi-modal action distributions, thus enhancing exploration strategies. By applying AM to both PPO and SAC, which represent on-policy and off-policy RL algorithms respectively, the authors provide comparative experimental results. These findings demonstrate that the AM method can transform state-wise constrained Markov Decision Processes (SCMDP) into Markov Decision Processes (MDP), thereby enhancing the sample efficiency of the original algorithms.

**Strengths:**

1. The motivation for this work is clear: it addresses the SCMDP problem by utilizing a model to learn a feasible action space, effectively converting SCMDP into an MDP and improving the algorithm’s sample efficiency.
2. The paper provides a step-by-step explanation of related concepts, making it very accessible to readers who may not be familiar with the field.
3. The description of the action and state spaces for the tasks in the experiments is clear, and the importance of the AM algorithm is effectively illustrated through visualizations at the end of the experimental section.

**Weaknesses:**

1. The absence of accompanying code makes it difficult to replicate the experimental results.
2. The experimental section appears somewhat limited, as it only tests the method in two environments. This reduces the persuasive power and credibility of the results.
3. The action mapping approach is not end-to-end, requiring pre-training with trajectory data before its application. This introduces additional costs, which are not adequately discussed in the paper.

**Questions:**

1. The paper's theme is closely related to Safe RL, as mentioned. However, many Safe RL algorithms are not included as baselines for comparison. What is the rationale behind this omission?
2. While PPO+Replacement has slower learning efficiency, it strictly ensures the satisfaction of constraints. This property is valuable in some online environments, but AM lacks this capability. Are there any proposed methods to address this limitation?
3. In Figure 4(c), the performance of AM-SAC suddenly increases at 7500 steps. How can this phenomenon be explained?

---

> ### Author Response · Authors · 2024-11-28
> **Answer to Questions**
>
> Thank you very much for your feedback. We modified the manuscript to address your questions and concerns.
>
> ### To Weakness 1:
> We added Algorithms 1+2 to describe the training process of action mapping and will release the code if or when it gets published.
>
> ### To Weakness 2:
> Unfortunately, we could not add more environments, but we hope the other added experiments and ablations help strengthen the case for action mapping.
>
> ### To Weakness 3:
> Action mapping does not require trajectory data for pretraining. We now elaborate on how the feasibility policy is trained in Appendix A. The added computational cost was discussed in Appendix F, showing that pretraining requires 5.5h / 16.8h for the robotic arm environment and 3.3h / 16.3h for the path planning environment, with the second number being the overall training time.
>
> ### Question 1:
> We originally thought a comparison with model-free approaches is unfair since they do not have access to the inductive bias of the feasibility model. However, we added a Lagrangian PPO and SAC implementation to the comparison. They do improve constraint satisfaction slightly compared to PPO and SAC but do not help or even hamper performance. Overall, action mapping and action projection perform significantly better.
>
> ### Question 2:
> It is indeed a valuable property if it is achievable. Since replacement only works if a feasible replacement action is known, it is not generally applicable, such as in the path planning environment. Where applicable, it can, in principle, be added to any algorithm.
> Therefore, in the robotic arm environment, we now also combined AM-PPO with replacement, which yields even better performance with guaranteed constraint satisfaction.
>
> ### Question 3:
> We had a discussion on this in the paper, but we added some text to make it more clear:
> “Additionally, in contrast to all other approaches, the action mapping agent has a second jump in performance and constraint satisfaction between 5M and 10M steps. This leap can be attributed to the objective policy’s understanding of the obstacles and incorporating them into the plan instead of only being nudged around by the feasibility policy.”

---

### Author Response · Authors · 2024-11-28
**List of Changes**

We would like to thank the reviewers for their constructive feedback. We tried to address all of the questions and concerns and believe it made the paper stronger. We hope the reviewers have the time to check the updated manuscript to see whether their concerns were sufficiently addressed.

List of major changes (chronologically):
- Emphasize the difference to Theile et al. (2024)
- Discussed the action representation literature
- A thorough description of the objective policy and the overall training procedure is added in Section 4.2 and Algorithms 1+2
- A comparison with Lagrangian approaches is added
- Combined AM-PPO and action replacement in the robotic arm environment
- Discussed the assumption of having a feasibility model in the Conclusion
- Provided a detailed description of the feasibility policy training in DRL environments in Appendix A
- Added the network architecture and hyperparameters in Appendix B
- Added a description of the Lagrangian algorithms in Appendix C.2
- Added a study on the approximation accuracy of the feasibility model in the path planning environment and its impact on learning performance in Appendix G

---

### Meta-Review · Area_Chair_455z · 2024-12-20

**Metareview:**

This paper presents a novel method for training a feasibility policy in safe reinforcement learning, demonstrating promising results in terms of constraint violation reduction and improved returns.  However, the submission suffers from several weaknesses that prevent its acceptance in its current form. While the core idea is interesting and the approach is conceptually straightforward, the assumption of a pretrainable feasibility policy appears overly restrictive and lacks generalizability. Furthermore, the experimental evaluation is limited in scope, both in terms of environments and baseline comparisons. Expanding the experiments to include more diverse and challenging constrained RL environments, along with a broader range of baseline algorithms (including adapted Lagrangian methods), would significantly strengthen the paper.  Given these limitations, particularly the strong assumption and limited empirical validation, we reject this submission but encourage the authors to address these concerns and resubmit a revised version to a future venue.

**Additional Comments On Reviewer Discussion:**

Some concerns from the reviewers are addressed, however it is still insufficient to make the cut.

---

### Decision · Program_Chairs · 2025-01-22

Reject